# Preparation and Study of Solid Lipid Nanoparticles Based on Curcumin, Resveratrol and Capsaicin Containing Linolenic Acid

**DOI:** 10.3390/pharmaceutics14081593

**Published:** 2022-07-30

**Authors:** Roberta Cassano, Simona Serini, Federica Curcio, Sonia Trombino, Gabriella Calviello

**Affiliations:** 1Dipartimento di Farmacia, Salute e Scienze della Nutrizione, Università della Calabria, Arcavacata di Rende, 87036 Cosenza, Italy; roberta.cassano@unical.it; 2Dipartimento di Medicina e Chirurgia Traslazionale, Sezione di Patologia Generale, Facoltà di Medicina e Chirurgia, Università Cattolica del Sacro Cuore, Largo F. Vito, 00168 Roma, Italy; simona.serini@unicatt.it (S.S.); gabriella.calviello@unicatt.it (G.C.); 3Fondazione Policlinico Universitario A. Gemelli IRCCS, Largo F. Vito, 00168 Roma, Italy

**Keywords:** linolenic acid, curcumin, resveratrol, capsaicin, solid lipid nanoparticle, atopic dermatitis

## Abstract

Linolenic acid (LNA) is the most highly consumed polyunsaturated fatty acid found in the human diet. It possesses anti-inflammatory effects and the ability to reverse skin-related disorders related to its deficiency. The purpose of this work was to encapsulate LNA in solid lipid nanoparticles (SLNs) based on curcumin, resveratrol and capsaicin for the treatment of atopic dermatitis. These compounds were first esterified with oleic acid to obtain two moonoleate and one oleate ester, then they were used for SLN matrix realization through the emulsification method. The intermediates of the esterification reaction were characterized by FT-IR and ^1^N-MR analysis. SLNs were characterized by dimensional analysis and encapsulation efficiency. Skin permeation studies, antioxidant and anti-inflammatory activities were evaluated. LNA was released over 24 h from nanoparticles, and resveratrol monooleate-filled SLNs exhibited a good antioxidant activity. The curcumin-based SLNs loaded or not with LNA did not induce significant cytotoxicity in NCTC 2544 and THP-1 cells. Moreover, these SLNs loaded with LNA inhibited the production of IL-6 in NCTC 2544 cells. Overall, our data demonstrate that the synthesized SLNs could represent an efficacious way to deliver LNA to skin cells and to preserve the anti-inflammatory properties of LNA for the topical adjuvant treatment of atopic dermatitis.

## 1. Introduction

Atopic dermatitis is a chronic, relapsing and remitting inflammatory skin disease caused by a complex interaction of immune dysregulation, epidermal genetic mutations and environmental factors that destroy the epidermis, resulting in intensely itchy skin lesions with papules located mainly on the face, elbows, friction zones and flexural areas [1]. Maintenance therapy consists of the use of emollients, daily bathing with soap-free detergents, wet wrap therapy and phototherapy [2]. At skin level, the topical route of drug administration is very advantageous as compared to the systemic one, since it allows the drug to be applied directly to the intended site of action, preventing it from entering the bloodstream [3]. For instance, among the first-line pharmacological approaches, there is the topical application of immunosuppressors such as corticosteroids to reduce inflammation and calcineurin inhibitors for the therapy of eczema. Moreover, antibiotics are widely used topically to eradicate skin infections, as well as antihistamines to relieve itchy skin. 

However, the topical drugs have low efficiency and specificity of action related to poor penetration through the epidermal layer of the skin. For this reason, they could cause skin atrophy, burning sensations and systemic side effects leading to poor patient compliance. In this contest, nanoformulations are considered very promising drug delivery carriers for the topical administration of drugs deep into the epidermis, allowing better drug release profiles to achieve therapeutic goals [4]. Particular attention is being paid to the use of plant-derived non-steroidal anti-inflammatory agents, vitamins, fatty acids and minerals useful as lipid matrix components or incapsulated agents, since they can improve skin inflammation and reduce the use of corticosteroids [5]. Linoleic acid (LA, C18:2 n-6) and α-linolenic acid (LNA, C18:3 n-3) are the two main essential polyunsaturated fatty acids (PUFAs) present in the *Stratum corneum* lipid layer, which is crucial for the barrier integrity [6]. In fact, they have a protective function and influence skin structure and physiology. In particular, LNA can be metabolized to the long chain PUFAs eicosapentaenoic acid (EPA, 20:5 n-3), docosahexaenoic acid (DHA, 22:6 n-3) and their metabolic derivatives resolvins, which modulate the response of the cutaneous cells to various inflammatory and immunogenic “*stimuli*” [7]. Recent studies have also demonstrated that LNA can reduce skin itching in animal models of atopic dermatitis [8]. In addition to LNA, other natural compounds are also used to treat skin alterations [9]. Over the last decades, it was also recognized that LNA may also exert anti-oxidant activity in different experimental conditions [10]. In particular, LNA was found to inhibit ROS production in endothelial cells isolated from rats and treated with the antineoplastic drug chlorambucil [11]. Moreover, it has been demonstrated that LNA reduced the formation of lipid peroxidation products (such as MDA) in human lymphocytes stimulated with phytohemagglutinin A [12]. The authors suggested that such an effect was related to the induction of the antioxidant enzymes catalase (CAT) and superoxide-dismutase (SOD). This hypothesis was also put forward by Raj et al., who found that LNA reduced MDA production though an increase of CAT and SOD activities in a rat model of intestinal toxicity induced methotrexate [13]. Finally, it was reported that LNA was able to reduce lipid peroxidation and induce the activities of SOD, CAT and glutathione peroxidase in rats treated with the pro-oxidant methyl mercury [14,15].

Curcumin (CU) is a bioactive polyphenol extracted by the spice turmeric, originating from *Curcuma longa*, and it is largely known for its marked anti-inflammatory activity [16].

Resveratrol (RV) is a phytoalexin present in almost 70 plant species and possesses anti-inflammatory, immunomodulatory and antioxidant activity [17]. In particular, the anti-inflammatory activity has been shown to be partly related to a relaxing effect on vessels causing an improvement in skin microcirculation. For this property, it is considered a valuable natural remedy for the treatment of irritant dermatitis [18].

Capsaicin is an alkaloid derived from the seed and membranes of plants of the nightshade family (active principle of chili pepper). The repeated application of topical capsaicin abolishes pain and itch [19]. In this contest, solid lipid nanoparticles (SLNs) have been showing promising results to enhance molecule penetration up to dermis, to reduce systemic absorption and to provide chemical stability of compounds sensitive to light oxidation and hydrolysis [20].

The present work aimed to the realization of SLNs based on CU, capsaicin and RV esterified with oleic acid (OA) for LNA delivery (Figure 1). Our objective was to create carriers capable of penetrating the *Stratum corneum* of the dermis to promote LNA release. These systems would limit the use of corticosteroids and the undesirable effects resulting from them.

## 2. Materials and Methods

### 2.1. Materials

The substances used for the three synthesis reactions were: capsaicin, CU, RV, OA, LNA (Sigma Aldrich, St. Louis, MI, USA, VWR Chemical Prolabo and Alfa Aesar, Haverhill, MA, USA), dicyclohexycarbodiimide (DCC), 4-dimethylaminopyridine (DMAP) (FlukaChemika-Biochemika, Buchs, Switzerland, CH), distilled water, sodium taurocholate hydrate (salt), butane-1-ol, tween-20, butyroxytoluene (BHT), hydrochloric acid (HCL), tertiary-butyl alcohol, tertiary-butyl alcohol (TBA), trichloroacetic acid (TCA) (Sigma Aldrich, Saint Louis, MO, USA). The solvents used were: tetrahydrofuran (THF), chloroform, methanol (CH_3_OH), ethanol (CH_3_CH_2_OH) (Sigma Aldrich, VWR Chemical Prolabo and Alfa Aesar).

### 2.2. Instruments

^1^H-NMR spectra were realized by using a Bruker VM 30 spectrophotometer (MI, IT), FT-IR spectra were obtained using a Jasco 4200 spectrophotometer (MI, IT). UV-Vis spectra were realized using a Jasco V-530 UV/Vis spectrophotometer, while SLN dimensional analyses were carried out by means of Brookhaven 90 Plus Particle Size Analyzer. Rotavapor were used to remove solvent; while freezing-drying Micro Modulyo, Edwards was used to freeze-dry samples. Thin-layer chromatography (TLC) was performed using silica gel plates 60 F254 on aluminum supplied by Merck (DE) using UV light at a wavelength of 254 nm.

### 2.3. Synthesis of Capsaicin, CU and RVesters with Oleic Acid

In three different flasks, under magnetic stirring, three esterification reactions were prepared. Here we briefly describe the reaction between capsaicin and oleic acid. For RV and CU the procedure is identical. To a solution of oleic acid in THF, DMAP and DCC were added and left under magnetic stirring for 10 min at 0 °C. Then a capsaicin solution was slowly added and left under magnetic stirring at room temperature for 24 h (see Table 1). After drying, the raw product was washed with hot methanol in order to remove the reaction by-product dicyclohexylurea.

The same procedure, with the same conditions, was carried out in the other two flasks for the esterification reactions of CU and RV [21]. The quantities used are reported in Table 1. Finally, all the reaction products were thoroughly purified by chromatography on Merk 60 silica gel flash column (70–230 mesh) using as eluents: CHCl_3_/n-hexane (8:2) for CU-monooleate, methanol/CHCl_3_ (7:3) for RV-monooleate and capsaicin oleates. The purified products were characterized by nuclear magnetic resonance (NMR) spectroscopy and infrared radiation (IR) spectroscopy.

### 2.4. SLN Preparation

SLNs were prepared through microemulsion technique according to Trombino et al. [22]. Briefly, CU-monooleate, RV-monooleate and capsaicin-oleates were melted at a temperature of about 70 °C, in the presence or not of LNA. Afterwards, to obtain an O/A microemulsion, a solution of Tween 20, sodium taurocholate and butanol in water was added to the esters and maintained under magnetic stirring in an ice bath for 45 min (Table 2).

Subsequently, the obtained dispersions were lyophilized [23,24].

### 2.5. Encapsulation Efficiency Determination

Encapsulation efficiency (EE%) is defined as the percentage of loaded drug respect to the total amount of the used drug, and is calculated through this Equation (1),
(1)$$\mathrm{EE}\%=\frac{gf}{gi}\times 100\;$$
where *gi* indicates the total amount of drug used and *gf* the loaded drug amount.

SLNs were solubilized in a water/methanol solution (1:9) and were sonicated for 15 min at 37 °C. The absorbance of the compounds was determined by UV-Vis spectrometry at wavelength for LNA (λ = 264 nm, ε = 17,596×).

The tests were carried out in triplicate and the results were in agreement with ±5% standard error.

### 2.6. Transmission Electron Microscopy (TEM)

The morphology of the nanoparticles was assessed by TEM. A small amount of solution, obtained by microemulsion, was applied to a carbon-coated copper grid and left for a few minutes in order to promote the adhesion of the nanoparticles to the carbon substrate. The excess solution was removed by adsorption with a piece of filter paper. A drop of 1% phosphotungstic acid solution was applied and the air-dried sample was observed using a ZEISS EM 900 electron microscope at an accelerating voltage of 80 kV.

### 2.7. In Vitro Skin Penetration Studies

A Franz diffusion cell equipment was coated with skin withdrawn from rabbit ears (provided by the local butcher) and used for skin penetration experiments (*n* = 3), keeping the temperature in the range from 35.5 °C to 37 °C to mirror the physiological conditions. A sodium chloride (0.9%) solution (7 mL) containing ethanol (20%) was placed in the receptor chambers and was continuously stirred to obtain sink conditions. Ethanol used as a co-solvent increased the water solubility of the LNA. At specific time intervals, an aliquot (7 mL) of each sample was taken from the receptor chambers and replaced with fresh release medium. Samples were analyzed by UV-Vis spectrophotometry and drug release profiles were described as the percentage of drug released relative to the total amount loaded as a function of time.

### 2.8. Evaluation of Antioxidant Activity

The antioxidant activity was evaluated through malondialdehyde (MDA) test. 

An ethanolic solution containing 0.2% BHT (0.07 mL), 0.5% TCA (3 mL) and of TBA (0.5 mL) was prepared, and 1 mL of microsomal suspension was added to it. The samples were then incubated in a thermostatic bath at 37 °C, and they were withdrawn at 1 h, 3 h, 6 h and 24 h. Then, they were heated at 90 °C and centrifuged. The levels of TBA-MDA complex were evaluated spectrophotometrically (λ = 533 nm).

### 2.9. Cell Lines

In order to evaluate the cytotoxic and the anti-inflammatory effect of the CU- and RV-based SLNs (containing or not LNA), we treated two different cell types, the immortalized human NCTC 2544 keratinocytes and the THP-1 human monocytic leukemia cells, both cell types extensively used to mimic the cells normally present in the skin and both involved in the modulation of the inflammatory process underlying atopic dermatitis.

NCTC 2544 cells (a kind gift from Dr. R. De Bellis, Urbino, Italy) were maintained in Dulbecco’s modified minimal essential medium (DMEM) containing 2 mM glutamine, antibiotics (100 U/mL penicillin, 100 µg/mL streptomycin) and 10% fetal bovine serum (FBS). THP-1 monocytes were purchased from the American Type Culture Collection (ATCC, Manassas, VA, USA) and were maintained in RPMI 1640 culture medium containing glutamine (2 mM) and 10%FBS. The differentiation of THP-1 cells into M2 macrophages was obtained by exposing the cells to 320 nm phorbol-12-myristate acetate (PMA, Sigma-Aldrich, St. Louis, MO, USA) for 24 h.

### 2.10. MTT Assay

The MTT assay was used to measure cellular metabolic activity as an indicator of cell viability, proliferation and cytotoxicity. This colorimetric assay is based on the reduction of a yellow tetrazolium salt 3-(4,5-dimethylthiazol-2-yl)-2,5-diphenyltetrazolium bromide (or MTT) to purple formazan crystals by metabolically active cells. The viable cells contain the mitochondrial succinate dehydrogenase enzyme, which transforms the tetrazolium ring of MTT (yellow) to formazan (blue crystals). The insoluble formazan crystals are dissolved using a solubilization solution and the resulting colored solution is quantified by measuring absorbance at 570 nanometers using a multi-well spectrophotometer. The darker the solution, the greater the number of viable, metabolically active cells. NCTC 2544 and THP-1 cells were seeded at the concentration of 5 × 10^3^ cells/well in a 96-well multiwell culture plate in a final volume of 200 µL/well. THP-1 cells were treated with 320 nM PMA to induce their differentiation into M2 macrophages (cell differentiation was observed microscopically as a change in their morphology and their adhesion to the bottom of the wells). After 24 h, cell culture medium was removed and replaced with fresh culture medium containing or not empty SLNs (containing CU or RV, but not LNA) and SLNs loaded with LNA at the concentrations of 0.5, 1.0 and 5.0 µg/mL (from stock solutions obtained by solubilizing the SLNs directly in the culture medium at the concentrations of 2 mg/mL). At the indicated time points (24, 48 and 72 h), 50 µL of MTT solution (2 mg/mL in phosphate buffer saline) were added into each well and the culture plate was further incubated at 37 °C for 4 h. Then, surnatant was removed and the formed crystals were solubilized by using dimethyl sulfoxide (DMSO, 100 µL/well). The absorbance was measured at 570 nm and 630 nm (as a basal absorbance to be subtracted to the absorbance at 570 nm) by using a plate spectrophotometer (Tecan, Männedorf, Switzerland). Cell viability was calculated by using the following formula:$$\%\;\mathrm{of}\;\mathrm{viable}\;\mathrm{cells}=\frac{{\mathrm{Absorbance}}_{\left(570-630\right)}\;\mathrm{treated}\;\mathrm{cells}}{{\mathrm{Absorbance}}_{(570-630)\;}\mathrm{control}\;\mathrm{cells}}\times 100\;$$

### 2.11. ELISA Analysis of IL-6 and MCP-1 Pro-Inflammatory Cytokines

In order to evaluate the anti-inflammatory activity of the CU- and RV-based SLNs, containing or not containing LNA, NCTC 2544 cells were seeded at the concentration of 250 × 10^3^ cells/well in a 12-well multiwell culture plate. After 24 h, cell culture medium was removed and replaced with fresh culture medium containing or not the empty SLNs (containing only CU or RV) and SLNs loaded with LNA (at a final concentration of 5 µg/mL) in the absence and in the presence of TNF-α (20 ng/mL), as a pro-inflammatory stimulus. At the end of the treatment period (24 h), surnatant was collected and stored at −80°C until the evaluation of the cytokine secretion. The presence of the pro-inflammatory cytokines IL-6 and MCP-1 were evaluated by ELISA assay using commercially available kits (ELISA MAX^TM^ Deluxe Set Human IL-6, cat. 430504; ELISA MAX^TM^ Deluxe Set Human MCP-1/CCL-2, cat. 428804, Bio-Legend, San Diego, CA, USA) following the instructions of the manufacturer. The minimum level of measurable cytokines was 4 pg/mL and 3.9 pg/mL, for IL-6 and MCP-1, respectively.

### 2.12. Statistical Analysis

Data were analyzed by unpaired *t*-test (Figures 5 and 7) and by the one-way analysis of variance (one-way ANOVA) followed by Tukey’s test (in Figures 8–13).

## 3. Results and Discussion

### 3.1. Esterification Reactions

The esterification reactions between CU/RV/capsaicin and OA were conducted to obtain esters to be used as a lipid matrix in SLNs to carry LNA. These syntheses were conducted in anhydrous tetrahydrofuran (THF) at room temperature (Figure 1).

The formation of the esters was confirmed by FT-IR and ^1^H NMR. In particular, the FT-IR spectrum of the CU-monooleate (Figure 2b), compared with those of the starting substances, reveals the presence of a new band at 1738 cm^−1^ that can be attributed to the stretching vibration of the C=O group of the ester. In addition, both the typical bands of the stretching vibration of aromatic CHs between 3110 and 3025 cm^−1^ and those of the phenolic OH at 3328 cm^−1^ can also be observed.

The FT-IR spectrum of capsaicin oleate (b) was also compared with that of capsaicin (c) and OA (a) (Figure 3b). The spectrum (b) shows the presence of anew band at 1763 cm^−1^ that can be attributed to the stretching vibration of the C=O group of the ester. In addition, both typical bands of the stretching vibration of the aromatic CHs between 3010 and 3067 cm^−1^ and those of amide NHs at 3316 cm^−1^ can also be observed.

Regarding the spectrum of RV monooleate (Figure 4b), this was also compared with that of OA (c) and RV (a) (Figure 4b). The spectrum (b) shows the presence of a new band at 1758 cm^−1^, which can be attributed to the stretching vibration of the C=O group of the ester. In addition, typical bands of the stretching vibration of aromatic CHs between 3121 and 3010 cm^−1^ are also observed.

The ester formation was also confirmed by ^1^H-NMR. In particular, the spectrum of the CU-monooleate showed the typical signals of both the protons of the aliphatic OA groups, and those of the phenolic rings of the CU. ^1^H NMR (DMSO-d6): δ 9.50 (1 H, s, OH), 7.60–6.20 (11 H, m, 6 H aromatic and 4 H ethylene, 1 H OH), 5.43 (2 H, m, ethylene CH_2_), 4.50 (1 H, m, CH), 3.90–3.81 (6 H, s, methoxyl CH_3_), 3.25 (2 H, m, CH_2_, methylene), 2.54–2.13 (6 H, m, methylene CH_2_)1.33–1.24 (22 H, m, methylene CH_2_), 0.86 (3 H, t, methylene CH_3_). Similarly, in the spectrum of capsaicin oleate, there are signal characteristics of the protons of the carbons of the OA, and those attributable to the protons present on the carbons of the aromatic ring of capsaicin. ^1^H NMR (DMSO-d6): δ 8.18 (1 H, s, NH), 7.20–6.78 (3 H, m, CH aromatics), 5.51–5.40 (4 H, m, ethylene CH_2_), 3.81 (3 H, s, methoxyl CH_3_), 4.24 (2 H, s, CH_2_ ethylene), 2.40–2.12 (11 H, m, CH and CH_2_ alkyl), 1.71–1.24 (26 H, m, CH_2_ methylene), 0.90–0.82 (9 H, m, methyl CH_3_). The spectrum of the RV-monooleate also confirmed the esterification by exhibiting both the signals related to the protons of the aromatic rings of RV, and those of the aliphatic chain of the OA chain. ^1^H NMR (DMSO-d6): δ 9.08 (2 H, s, OH), 7.80–7.30 (4 H, m, aromatic CH), 6.95–6.30 (5 H, m, 3 H aromatic and 2 H ethylene), 5.33 (2 H, m, ethylene CH_2_), 1.70–1.21 (28 H, m, methylene CH_2_), 0.85 (3 H, t, methylene CH_3_).

### 3.2. Preparation and Characterization of SLNs Based on CU-Monooleate, Capsaicin Oleate and RV-Monooleate

Empty SLNs and LNA-loaded SLNs were successfully prepared via the microemulsion technique with an encapsulation efficiency of 62%, 85% and 99% for CU-monooleate, capsaicin oleate and RV-monooleate, respectively. Dynamic light scattering analysis made it possible to determine the average diameter of the nanoparticles and their polydispersion index (PI), as shown in Table 3. These PI values are indicative of a good homogeneity in the distribution of the particle size.

### 3.3. Characterization of SLNs

The obtained SLNs were characterized by TEM micrographies. The results showed that SLNs based on CU-monooleate and RV-monooleate, containing linolenic acid, possess a spherical shape, with dimensions ranging from approximately 200 nm to 400 nm (Figure 5).

### 3.4. In Vitro Skin Permeation Studies

SLNs containing LNA were also subjected to transdermal release studies (Figure 5). The LNA release studies were performed on all prepared SLN types, by withdrawing the solution from the receptor compartment at regular time intervals (1 h, 3 h, 6 h, 24 h). The results obtained (Figure 6) revealed that LNA was released over the 24 h from the SLNs in amounts ranging from 57% to 79% of the total amount loaded (21 mg). It can be observed that the highest release of LNA was obtained from the RV-based SLNs. Instead, a lower and comparable LNA release was observed in capsaicin- and CU-based.

### 3.5. Evaluation of Antioxidant Activity

The antioxidant ability of empty SLNs and LNA-loaded SLNs was evaluated in rat liver microsomal membranes over a 24 h incubation period by measuring the inhibition of lipid peroxidation induced by a free radical generator, such as *tert*- BOOH. From this point on, only the SLN based on CU-monooleate and R-monooleate were analyzed due to the irritating capacity of capsaicin. All the nanomaterials were able to preserve the antioxidant capacity of the precursors (Figure 7). In particular, the most powerful antioxidant activity was exhibited by the RV-based SLNs. Since LNA, due to its chemical structure, is subject to peroxidation, the results obtained suggest that RV may be able to better protect the fatty acid and the membranes that become enriched in it from oxidative insults with respect to CU. On the other hand, from the data it appears that CU is able to preserve LNA from oxidation only until the first three hours of incubation. Later, this effect is lost, and, in the presence of LNA, a higher production of MDA is observed, probably due to a more precocious consumption of CU as compared to RV in these experimental conditions.

### 3.6. Stability Study of Nanoparticles

The stability of SLNs was assessed at room temperature (25 °C) and refrigerated (4 °C) for 60 days. The formulations were analysed for size, PDI and EE. The results indicated that negligible changes were observed in all measured parameters, maintaining good stability for approximately two months.

### 3.7. Evaluation of Cell Viability by MTT Assay

The effects of CU- and RV-based SLNs containing or not LNA on the viability of the immortalized keratinocytes NCTC 2544 and of the THP-1 monocytes are shown in the Figure 8, Figure 9, Figure 10 and Figure 11. We observed that (Figure 8A) keratinocytes treated with empty CU-based SLNs show a viability higher than 95% even at the higher concentration used (5 µg/mL) and for the longer treatment period (72 h). It can be noticed that after 24 h cell viability was even higher than that observed in control cells, suggesting that these SLNs lack completely any cytotoxic effect in keratinocytes. However, the presence of LNA induced a cell viability reduction of about 32% as compared to untreated cells starting from 48 h treatment (Figure 8B). In THP-1 monocytes, both empty CU-based SLNs and LNA-loaded CU-based SLNs showed a time-dependent tendency to improve cell viability as compared to untreated cells (Figure 9A,B). This finding is of interest, since in our experimental model THP-1 monocytes were preliminarily treated with PMA, which induces their differentiation towards an anti-inflammatory and pro-resolving M2 macrophage phenotype. In fact, in a previous work of ours, we demonstrated that, after PMA treatment, THP-1 cells expressed high levels of CD206, a specific marker of M2 macrophage polarization [25]. In a chronic inflammatory setting, such as that of atopic dermatitis, an increased survival and metabolic activity of M2 macrophages, such as that shown by the MTT assay in the presence of CU-based SLNs, could help the resolution and healing of the inflammatory process.

Similarly, the empty RV-based SLNs also did not reduce cell viability of NCTC 2544 cells at any concentration used and for all the treatment period (24–72 h) (Figure 10A). On the other hand, in the presence of LNA, the RV-based SLNs induced a decrease of about 30% of cell viability (Figure 10B) as compared to untreated cells after 48 h treatment. It can be noticed, however, that cell viability values reverted to those of untreated cells after 72 h. This effect was probably due to the fact that, after this long period, the LNA released from RV-based SLNs could be completely consumed, as suggested by the faster release of LNA from these SLNs, as compared to that released from CU-based SLNs (see Figure 6).

In THP-1 cells, empty RV-SLNs did not affect cell viability at any concentrations used until 24 h, whereas a slight decrease (never higher than 20%) was observed by prolonging the treatment period (48–72 h) (Figure 11A). In the presence of LNA, the cell viability of THP-1 cells was reduced to about 25% in all the concentrations used for the first 48 h, but after 72 h treatment cell viability values even increased as compared to untreated cells. This finding confirms what was also observed in NCTC 2544 cells (Figure 11B).

### 3.8. Evaluation of Anti-Inflammatory Activity

Figure 12 shows the ability of the CU- and RV-based SLNs (panels A and B, respectively) containing or not containing LNA in inhibiting the production of the pro-inflammatory cytokine IL-6, both in basal conditions and in the presence of the pro-inflammatory stimulus TNF-α. In particular, we chose to study this cytokine since it is one of the most important inflammatory mediators involved in the development of atopic dermatitis. We observed that, while empty SLNs, containing only CU (CU-SLNs) or only RV (RV-SLNs) did not exert any inhibitory effect on the production of IL-6 in basal conditions, the CU-based SLNs containing LNA (CU-LNA-SLNs) (Figure 12A) were able to significantly reduce the production of the pro-inflammatory cytokine (inhibition of 14.4% vs. Ctrl, *p <* 0.05). This result indicates that the addition of LNA to CU-SLNs could represent an effective strategy to confer an anti-inflammatory potential to these antioxidant nanoparticles.

On the other hand, the RV-LNA-SLNs(Figure 12B) not only did not inhibit IL-6 production in basal conditions, but even significantly increased it (increase of 26.4% vs. Ctrl, *p <* 0.05). This finding agrees with what we previously observed in the cell viability experiments, where the RV-based SLNs released LNA faster than the CU-based ones (within 3 h). It is possible that the fast-released LNA, not any more protected by the RV antioxidant activity, may easily undergo oxidative deterioration, thus not allowing its anti-inflammatory activity to be exerted.

The addition of TNF-α to the culture medium induced a huge increase in the secretion of IL-6 by NCTC 2544 cells (Panel A: 197% increase vs. Ctrl; Panel B: 219% increase vs. Ctrl). Interestingly, the treatment with CU-SLNs reduced conspicuously the IL-6 production (Reduction: 94.1% vs. TNF-α treated cells, *p* < 0.05). The addition of CU-LNA-SLNs was able to further reduce IL-6 secretion (by 95.5%, vs. TNF-α, Figure 12A). This finding suggests that the synthesized nanomaterials could exert their maximal anti-inflammatory activity in the setting of an overt pro-inflammatory condition.

It is worth noting that in the presence of TNF-α, the RV-SLNs (Figure 12B) were also able to significantly reduce the production of IL-6 (reduction: 18% and 10% using RV-SLNs and RV-LNA-SLNs, respectively, *p <* 0.05), even if at a lower extent than the CU-based SLNs. Again, this finding suggests that the early release of LNA from these SLNs may reduce its anti-inflammatory effect. Nonetheless, the results obtained indicate that the RV-based formulations, possessing considerable antioxidant efficacy (see Figure 7), may also promote the reduction of the pro-inflammatory process in atopic dermatitis.

Since it is known that the chemokine MCP-1 plays a crucial role in the recruitment of monocytes in skin chronic inflammatory disorders, including atopic dermatitis, we also analyzed the possible effects of the synthesized SLNs on the secretion of this cytokine in NCTC 2544 keratinocytes, both in basal conditions and in the presence of the pro-inflammatory cytokine TNF-α (Figure 13). We observed that in basal conditions, where the levels of MCP-1 secreted are very low (about 36 pg/mL), not one of the SLNs (empty or LNA-loaded) were able to modify MCP-1 production. However, in the presence of the pro-inflammatory stimulus TNF-α, which induces a considerable increase in the cytokine production (panel A, 491.6% increase; panel B: 482.4% increase; *p <* 0.05), a significant reduction of MCP-1 was found by using all the synthesized SLNs. In particular, we found that both CU-SLNs and CU-LNA-SLNs reduced at a very high extent (to nearly undetectable levels, significantly lower than those observed in untreated cells) the MCP-1 production (reduction 77.5% and 86.7%, respectively, *p <* 0.05 with respect to the TNF-α-treated cells). Similarly, RV-SLNs and RV-LNA-SLNs did not modify the basal production of MCP-1 (Figure 13B). However, RV-SLNs induced a significant decrease of TNF-α-induced MCP-1 secretion (163.6% reduction, *p <* 0.05, as compared to the TNF-α-treated cells). Of note, RV-LNA-SLNs induced a significant and an even more conspicuous decrease of it (425.3% reduction, *p <* 0.05 as compared to TNF-α-treated cells), demonstrating that the inclusion of LNA in the RV-based SLNs could improve the anti-inflammatory efficacy of empty RV-SLNs.

## 4. Conclusions

The aim of this work was the design, manufacture, and study of SLNs based on CU, RV, capsaicin and OA, potentially useful as delivery systems for LNA. SLNs were made from the respective esters (CU-monooleate/capsaicin oleate/RV-monooleate) through microemulsion technique. Nanoparticles showed good encapsulation efficiency, stability, and a size suitable for topical administration. This hypothesis is supported by the results obtained from transdermal release studies, which revealed that a variable cumulative percentage of NLA ranging from 57 to 79% of the total (21 mg) is released. Both the CU-based and the RV-based SLNs did not exert significant cytotoxic effects in both NCTC 2544 and THP-1 monocytes differentiated into M2 macrophages, and, in some cases, even increased it. Moreover, we found that the CU-based SLNs loaded with LNA were able to significantly reduce the production of the pro-inflammatory cytokine IL-6, both in basal conditions and in the presence of a pro-inflammatory stimulus (TNF-α). Even if the RV-based SLNs (both loaded or not with LNA) were unable to exert an anti-inflammatory effect in basal conditions, they significantly reduced the production of IL-6 in the presence of TNF-α. Of interest, the RV-LNA-SLNs were able to inhibit significantly more than the empty RV-SLN the production of MCP-1, a crucial cytokine in the recruitment of monocytes in the setting of skin chronic inflammatory disorders.

These data allow us to hypothesize a potential use of CU- and RV-based SLNs loaded with LNA for a more efficient delivery of this fatty acid to skin cells, as well as for the preservation of its anti-inflammatory activity for the topical adjuvant treatment of atopic dermatitis.

## Data Availability

Data are contained within the article.

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
