# Peer review of "Preparation and Study of Solid Lipid Nanoparticles Based on Curcumin, Resveratrol and Capsaicin Containing Linolenic Acid"

_pharmaceutics, 2022, doi:10.3390/pharmaceutics14081593_

Round 1
Reviewer 1 Report
The topic of this manuscript is interesting and fits well the scope of the journal. However, extensive amendments are still required.
1) Why the anti-inflammatory experiments were not carried out with positive control?
2) For anti-inflammatory experiments, did the authors use any stimulus to stimulate the secretion of IL-6?
3) Why IL-6 was chosen as inflammatory biomarkers bit not TNF-alpha?
4) How good is the stability of the SLNs?
Reviewer 2 Report
The manuscript describes the synthesis of curcumin and resveratrol esters with linolenic acid. The esters were formulated in solid lipid nanoparticles and the formulations evaluated for cytotoxicity against cells in culture and production of interleukin-6 (an inflammatory cytokine).
The manuscript communicates poorly the results and conclusions requiring a complete revision to become reproducible by others, readable and understandable. Below are some suggestions for revision.
1)Line 11, do not use undefined abbreviations in the abstract (e.g., PUFA)
2)Give the precise composition and concentrations of all components of the formulations and decide whether the most abundant component is indeed acid linolenic. Is acid linolenic the drug or the carrier? Do the SLN mostly contain LA and LA esters? Which are the emulsifiers of the SLN? Could you draw a representation of the SLN carrying the active compounds?
3)Line 39, replace "allows" by "allowing".
4)Line 44, replace "contest "by "context".
5)Multiple instances of joined words indeed prevent the reader from reading, e.g. "ourobjective" on line 85
6) On section 2.4., please give final concentrations of emulsifiers on lines 131, 132 or quote Table 2. Give also the final volume on Table 2.
7)On Table 2, replace "oasters "by 'esters "
8)One would not be able to reproduce the synthesis form the description on item 2.3. Furhtermore, all abbreviations on captions for Tables should be explicitly defined. The same holds for all figures and figure s captions.
9)Figure 6: it is not possible to understand the experiment. What means empty??? Whiout linolenic acid? More details are required in the figure caption. In addition, loaded and unloaded SLN gave similar inhibition of lipoperoxidation. Discuss the meaning of this result.
10)Figure 11, inhibition of IL-6 production was very similar for all SLN, empty or loaded. Discuss this.
11) Please, correct y-axis on Fugure 11 to "% inhibition of IL-6 production".
12) No additional effect of LA could be observed for SLN based on curcumin (CU) or resveratrol (RV) esters. The general interpretation is possibly wrong since effects were due to CU and RV esters.
10)
3)
Reviewer 3 Report
This manuscript reported the preparation of lipid nanoparticles based on curcumin, resveratrol and capsaicin derivatives. These nanoparticles were applied for antioxidant and anti-inflammation studies. The work is well designed, but some critical data are missing. I recommend major revision.
1. The 1H NMR spectra of synthesized curcumin, resveratrol and capsaicin derivatives should be presented in the manuscript.
2. Additional characterizations for nanoparticles should be conducted. TEM images of these nanoparticles should be captured to see the morphology of these nanoparticles.
3. The stability of these nanoparticles should also be evaluated during long-term storage.
4. Some mistakes in the manuscript should be corrected. e.g. In Table 3, "Polidispersion index" should be "Polydispersion index".
Round 2
Reviewer 1 Report
The manuscript has been improved and appears to be acceptable.
Reviewer 2 Report
The Italian on Figures should be translated into English. Minor corrections are still required before acceptance for publication.
Reviewer 3 Report
The authors have made adequate revisions. I only have one additional suggestion to the manuscript. I think the manuscript can be accepted after addressing the following issue.
1. The scale bars of TEM images should be presented in the figure.